# Hypothesis: Single Actomyosin Properties Account for Ensemble Behavior in Active Muscle Shortening and Isometric Contraction

**DOI:** 10.3390/ijms21218399

**Published:** 2020-11-09

**Authors:** Alf Månsson

**Affiliations:** Department of Chemistry and Biomedical Sciences, Linnaeus University, Universitetskajen, 391 82 Kalmar, Sweden; alf.mansson@lnu.se; Tel.: +46-708866243

**Keywords:** myosin, actin, single molecules, ensembles, muscle contraction, sarcomeres

## Abstract

Muscle contraction results from cyclic interactions between myosin II motors and actin with two sets of proteins organized in overlapping thick and thin filaments, respectively, in a nearly crystalline lattice in a muscle sarcomere. However, a sarcomere contains a huge number of other proteins, some with important roles in muscle contraction. In particular, these include thin filament proteins, troponin and tropomyosin; thick filament proteins, myosin binding protein C; and the elastic protein, titin, that connects the thin and thick filaments. Furthermore, the order and 3D organization of the myofilament lattice may be important per se for contractile function. It is possible to model muscle contraction based on actin and myosin alone with properties derived in studies using single molecules and biochemical solution kinetics. It is also possible to reproduce several features of muscle contraction in experiments using only isolated actin and myosin, arguing against the importance of order and accessory proteins. Therefore, in this paper, it is hypothesized that “single molecule actomyosin properties account for the contractile properties of a half sarcomere during shortening and isometric contraction at almost saturating Ca concentrations”. In this paper, existing evidence for and against this hypothesis is reviewed and new modeling results to support the arguments are presented. Finally, further experimental tests are proposed, which if they corroborate, at least approximately, the hypothesis, should significantly benefit future effective analysis of a range of experimental studies, as well as drug discovery efforts.

## 1. Introduction

Contraction of striated muscle (heart and skeletal muscle) is the result of ATP driven interactions between the contractile proteins actin and myosin II (Figure 1) [1,2]. These proteins are incorporated into thin and thick filaments, respectively, in a highly ordered arrangement in muscle sarcomeres [3,4]. The ~2 µm long sarcomeres connect repetitively in series to form ~1 µm wide myofibrils (Figure 1A) that run the length of a muscle. A muscle cell (muscle fiber) is packed with such myofibrils, connected in parallel with the sarcomere pattern roughly in register between neighboring myofibrils. This structural arrangement is evolutionarily optimized for its purpose, as evidenced by convergent evolution with a similar structure evolving independently in bilaterians (e.g., vertebrates) and non-bilaterian eumetazoans (e.g., jellyfish) [5]. Despite the high degree of order, the structure and function of a sarcomere are actually quite complex with a plethora of proteins, accessory to actin and myosin, with functions in the sarcomere assembly during development, structural integrity, signaling, and with modulatory/regulatory roles in muscle contraction [3,4,6]. Some of the accessory proteins are likely to have prominent roles in the latter regard, for example, as suggested by the prevalence of diseases associated with modifications (e.g., due to mutations) in these proteins [7,8]. In particular, this includes the regulatory thin filament proteins, troponin and tropomyosin [9,10]; the thick filament protein, myosin binding protein C (MyBP-C) [11]; and the large elastic protein, titin [12] that connects the thin and thick filaments. In accordance with the mentioned disease-associated proteins, they have also been found to affect several aspects of steady-state muscle contractions even at nearly full activation (see references below). Furthermore, in addition to the accessory proteins, the three-dimensional (3D) organization of the thin and thick filaments in the sarcomere (cf. [13]) may play important roles in sarcomere contractile function [14,15,16,17,18,19]. 

The functional and structural complexity increase further when considering thousands of sarcomeres in series and in parallel in a muscle cell (cf. Figure 1A) where force and shortening are developed by the simultaneous action of billions of myosin motors [22,23]. Whereas the ordered arrangement, i.e., with sarcomeres coupled in series along myofibrils and in parallel over the muscle cross-section, allows interpretation of contractile phenomena observed in muscle cells and myofibrils in terms of individual actin–myosin interaction phenomena, it can also introduce pitfalls in data interpretation. For instance, half sarcomeres in series along a myofibril may not have identical properties, for example, due to a slightly varying overlap between thin and thick filaments on average [20,24], expression of different protein isoforms [25,26], or other factors such as posttranslational modifications and varying activation levels [27,28].

Despite the apparent complexity and added functions of a muscle sarcomere as compared with isolated actin and myosin, recent modeling work [23,29,30,31,32] has successfully accounted for the contractile properties of muscle, for example, the force-velocity (FV) relationship (Figure 2) based on the single molecule properties of actin and myosin alone. The ensemble mechanokinetic models used for this purpose, were based on the theoretical formalism of Hill [33] ([23] for more intuitive description) and have been further defined, in terms of molecular properties, by kinetic schemes and free energy diagrams for different states (cf. Figure 3) using parameter values (and functions) from the literature (cf. Appendix A of this paper for similar model used here). Steady-state contractile properties of the huge ensembles of actomyosin motors in muscle are, then, derived by solving ordinary differential equations in the state probabilities with observable variables (force, ATP turnover rate, etc.) obtained from these probabilities. Transient contractile properties of muscle, are, instead, more easily derived using Monte Carlo simulations (as described in Methods, see also [31]). In the case of the models in [30,32], there is a 1:1 correspondence in the model cross-bridge states with the model in Figure 3 used in the present paper, whereas other previous models were simpler with the model states, AMD_PP_ and AMD_PiR_ (shown in Figure 3) merged into one state. The introduction of both the AMD_PP_ and the AMD_PiR_ state was required to account for effects of the small molecular substance blebbistatin. The existence of these states has also been suggested by recent structural and reverse genetics analysis of isolated myosin [34]. The parameter values used in previous modeling efforts were primarily derived from single molecule (particularly optical tweezers) studies and biochemical solution studies of isolated actin and myosin. Therefore, I denote them as bottom-up types of models [23,35], using terminology frequently adopted in fields such as information processing, management, nanotechnology etc., where a bottom-up approach denotes the piecing together of components (in this case single molecules) into more complex systems (half sarcomeres). This is opposed to a top-down approach which infers the opposite process of decomposition of the complex system. The bottom-up models, without adjustments of the preset parameter values more than within experimental uncertainties to fit the data, have been successful in accounting for the rate of increase in isometric force, at least for the upper 50% of the rising phase, for the steady-state FV relationship for shortening, and for several other features such as the relationship between sliding velocity and [MgATP] [29,31,32]. This was achieved without invoking effects of 3D order, accessory proteins (titin, troponin/tropomyosin, myosin-binding protein C), or emergent cooperative phenomena. These results imply that there are only minor, if any, influences of the listed properties on the studied muscle function. This view also gains support from experimental studies of the FV relationship using isolated actin-myosin ensembles [22,36,37]. These studies demonstrated an FV relationship for shortening at mM [MgATP] that was quite similar to that observed from muscle cells (Figure 2). The following hypothesis is stated based on the above findings:

**Hypothesis** **1.***The single molecule properties of actin and myosin account for ensemble contractile properties of a half sarcomere during shortening and isometric contraction at nearly saturating Ca concentrations*.


In this paper, first, the hypothesis is specified in greater detail with regard to scope and limitations. In this context, the meaning of bottom-up assignment of model parameter values based on single molecule properties is further specified. Then, different pieces of existing evidence that support or seem to falsify the hypothesis are considered. In this connection, new modeling data are presented and the modeling studies and actomyosin ensemble studies mentioned above are described in greater detail. In addition, previously available evidence is also presented. Finally, testable predictions following from the hypothesis are further considered and experimental systems to test these predictions are discussed in greater detail.

## 2. Methods

The key analyses, described below, relied on simulations using the model by Rhaman et al. [32], updated as in [29,30]. A kinetic scheme and free energy diagrams for the model are depicted in Figure 3. The origin of the parameter values in experimental studies are given in Appendix A in Appendix A as further considered in the Supplementary Text. 

Observable contractile properties were derived in the simulations from the populations of different cross-bridge states and their elastic properties, strain, etc. [33]. For example, force is given by the sum of the total number of attached cross-bridges multiplied by the stiffness and strain of each of these cross-bridges. The strain is given by the quantity x-x_i_., where x_i_ is the x-value for which the free energy of the given state has its minimum value (cf. Figure 3B showing x_1_, x_11_ and x_2_) and x is the coordinate describing the distance between a given point on the thick myosin filament and the nearest actin binding site. Here, this coordinate system is defined such that x = 0 nm, when the free energy in the AM state (Figure 3A) attains its minimum. In terms of the model, the free energy increases for x > x_i_ and x < x_i_ as a result of elastic energy added to chemical free energy of the specific state. In the case of linear cross-bridge elasticity with stiffness ks, the elastic contribution, G_E_, to the free energy, is given by G_E_ = ks(x-x_i_)^2^/2, i.e., the free energy diagrams vs. x have a parabolic shape (Figure 3B). In the simulations, the population of different states for each given x-value are either derived as average values by steady-state solutions of differential equations in state probabilities (cf. [32] and code in [29]) or as the absolute number of myosin heads in different states (associated with a specific strain) for each given point in time by the use of Monte Carlo simulations [31,32]. In the latter simulations, the waiting times between interstate transitions for each strain value (x) is calculated from the value of the rate functions (Figure 3A and Appendix A) using the Gillespie algorithm [41], starting with all myosin heads in the MDP state at the onset of the simulation (see further [31,32]). Specifically, here, average half sarcomere properties are simulated by assuming that 556 myosin-binding sites on actin are available for binding of myosin heads. This number corresponds to the total number of myosin binding sites along a 20 µm long filament on the assumption of 36 nm distance between neighboring sites, i.e., 20,000 nm/36 nm ≈ 556. This approach relies on assumptions that are intimately related to the idea that muscle function can be predicted from single actomyosin properties, as follows: (1) The muscle half-sarcomere contractile properties can be approximated by the contractile properties of approximately 20 actin filaments interacting with 10 thick myosin filaments. (2) This situation can be mimicked by surface adsorbed myosin motors interacting with a sufficiently long single actin filament (here taken as 20 µm) in an in vitro motility assay configuration (with added possibility to vary load on the filaments or velocity) with saturating surface density of 5000 active myosin motors per µm^2^.

These assumptions are supported by the following lines of argumentation: First, if we consider twenty 1 µm long (thin) actin filaments (with 556 binding sites for myosin in total) interacting with ten half (thick) myosin filaments that each contains 294 myosin heads, there would be 147 heads per thin (actin) filament. This is very similar to what is achieved in the in vitro motility assay approximation used in the simulations. Thus, with a surface density of 5000 active myosin heads per µm^2^ [42,43], a 20 µm long filament and a band of 30 nm [43] width, around the filament where surface-adsorbed myosin heads may reach actin, one arrives at a total number of heads, i.e., 5000 × 0.03 × 20 = 3000 per 20 µm of an actin filament, i.e., 150/µm. Second, if there is no cooperativity or emergent effects arising from the sarcomere arrangement (key elements of the hypothesis), the force and velocity is the same for a long single filament with a uniform distribution of myosin heads relative to the binding site as for twenty single actin filaments of approximately 1 µm length, interacting with a similar number of uniformly distributed myosin motors. Third, it is a reasonable assumption that the myosin motors in muscle are uniformly distributed relative to the nearest actin binding site as assumed in the in vitro motility assay-based Monte Carlo simulations. This idea was originally proposed by Huxley [44] based on the mismatch of the actin and myosin periodicities. Furthermore, more recently, further conditions such as variabilities in overlap and register between sarcomeres have been proposed to contribute to uniform distributions when considering a very large number of half sarcomeres working in parallel and series (reviewed in [23]). Naturally, this approximation is not valid when the interactions among a few (say less than 100 myosin motors and actin filaments) are considered. However, in principle, the assumption of uniform distributions also applies to Monte Carlo simulations of in vitro motility assays at both high and very low motor densities but, the exact positions of the limited number of myosin heads relative to the binding sites are determined by a random number generator. The latter distributes the individual heads in 360 bins, each of 0.1 nm width, with distances in the range 0–36/2 nm relative to a defined center of the nearest binding site. This approach allowed for simulations where the surface density of active myosin motors could be varied from very low to saturating density for simulating the properties of a muscle half sarcomere.

## 3. Results and Discussion

### 3.1. Detailed Specifications of Hypothesis and Bottom-Up Method of Assigning Model Parameter Values

The FV relationship for steady-state contractions, i.e., the relationship between the load on the muscle (equal to muscle force) and the shortening velocity, is central in model testing from Huxley [44] and onwards [45,46,47,48,49]. The fact that this relationship, at nearly full Ca^2+^-activation, is accounted for by the single molecule properties of actin and myosin, without the need to invoke any cooperative/emergent phenomena or any effect of 3D order and accessory proteins, is a central facet of the current hypothesis. The statement that there should be no effect of 3D order requires some clarification. This means that the order does not alter the actin–myosin interaction in a half sarcomere per se. However, for the effective operation of a whole muscle with billions of myosin heads and thousands of sarcomeres, in series and in parallel, the 3D order is presumably essential. This is both to ensure optimal packing of sarcomere proteins into the muscle, to preserve structural integrity during high-force contractions, and for effective summing of velocities/length changes along the muscle cell and forces over the muscle cross-section. 

The features of the FV relationship that are accounted for by single molecule properties include its general shape (and the related power output (force x velocity)) and energetics, the maximum velocity of shortening, and the maximum isometric force. It is hypothesized that the effects of drugs, altered concentrations of inorganic phosphate [Pi], [ATP], etc., on the steady-state FV relationship are accounted for. It is also hypothesized that some transient phenomena at close to saturating [Ca^2+^] are well approximated based on the single molecule properties. This includes the rate of increase in isometric force during a tetanus at tension levels >50% of the maximum force [50,51], the early phase of slow relaxation after an isometric tetanus [27,52], and, a slightly less exact (cf. [19]), the isometric tension transients in response to step changes in length, temperature, and [Pi]. However, the tension changes seen at low concentrations of Ca^2+^ (<80% of maximum) and tension levels <50% of the isometric value, are unlikely to be well accommodated (cf. [51]) based on only bottom-up actomyosin properties. This follows from data suggesting incomplete activation due to both thin [50] and thick [51] filament-based mechanisms at these tension levels. Furthermore, the response to a stretch of active muscle (eccentric contraction) involves additional complexities not seen during shortening and isometric contraction. This includes potentially emergent effects due to inter-sarcomere dynamics [53,54], effects due to stretching of titin [55,56], and finally, different kinetics of the actomyosin interaction than during shortening [57,58] such as slippage between neighboring sites [32] or inter-head cooperativity [59]. Therefore, the hypothesis is limited to isometric and shortening contractions at nearly maximal Ca^2+^ concentration (tentatively defined as corresponding to steady-state isometric tension >80%). Finally, whereas aspects of the hypothesis can, in principle, be tested using whole muscle cells and myofibrils, the hypothesis is stated for a half sarcomere of a myofibril. The reason is that complexities due to inter-sarcomere differences and various emergent phenomena may complicate results from cells and myofibrils.

If the hypothesis is correct, it should be possible to obtain model parameter values for prediction of the relevant phenomena from the bottom-up using isolated actin and myosin molecules (see further above). Such experiments that would provide parameter values include single molecule studies, for example, using optical tweezers to measure force and displacement, and experiments from disorganized ensembles e.g. biochemical solution kinetics to obtain zero-strain values for rate constants in the ATP driven actomyosin interaction mechanism. 

### 3.2. Existing Evidence in Favor of the Hypothesis

As briefly mentioned above, central pieces of evidence are the faithful prediction of a range of contractile data (e.g., the shape of the FV relationship, energetics, and rate of increase in isometric force) by models of the bottom-up type, as defined above [23,29,31,32] (see Introduction). These bottom-up models also predict the absolute values of the number of attached cross-bridges and isometric force per cross-sectional area. This, however, requires (in agreement with experimental findings) [60] the assumption of three myosin binding sites (three neighboring actin subunits) per properly oriented actin target zone, separated by ~36 nm along the thin filament [29]. In addition, the maximum velocity of shortening and the maximum power output are well predicted by such models [23,29,31,32]. Interestingly, potentially even better predictions for the maximum power and the maximum velocity of shortening are obtained if the cross-bridge elasticity is assumed to be nonlinear, as suggested by single molecule studies [29,61]. However, from reviews of primarily their own previous and new mechanical experiments on muscle cells, for example, stiffness measurements at different tension levels under rigor conditions (in the absence of MgATP), Linari et al. [62] claimed that they found evidence that cross-bridge stiffness was nearly linear in the myofilament lattice of muscle. Whereas I hold the experimental data by the researchers behind this paper [62] in high regard, I disagree with their interpretation. Thus, the evidence is inconclusive (e.g., compare with data from [61], e.g., Figures 3B and 5 in [61]) and insufficient as a basis for definite falsification of the idea of nonlinear cross-bridge elasticity in a muscle cell. More direct tests are required, as proposed below.

Further evidence that properties of single actomyosin interactions account for the ensemble FV relationship is provided by the similarity of experimental FV relationships recorded from living mammalian muscles and small isolated actomyosin ensembles, in the latter case without 3D order or accessory proteins. The latter type of studies have been performed by first adsorbing a small number of myosin motors (full length myosin or heavy meromyosin) to small silica beads [63] or optical fiber surfaces [36] (~<20 myosin heads) or by using myosin filaments (native thick filaments [22] (~<100 myosin heads) or myosin-rod cofilaments [37] (~<20 myosin heads)). Then, an actin filament held by optical traps [36,37] or by a cantilever [22] are brought in contact with the motor ensemble to obtain force and velocity data. Such studies were performed at close to physiological [MgATP] (>1 mM) and at temperatures >20 °C for the data illustrated in Figure 2 but similar results were derived previously in experiments at low [MgATP] (≤100 µM) [63]. A key feature of the data in Figure 2, is a similar maximum power output (maximum force x velocity) in muscle cells and isolated actomyosin ensembles, despite the lack of 3D order and accessory proteins in the latter case. If the latter factors have any key influence over contractile function in muscle cells, one would expect them to increase the maximum power output, which is a critically important aspect of muscle function. However, the data reproduced in Figure 2 suggest that the high maximum power output arises from properties inherent in the actomyosin interaction itself.

Similarities in steady-state properties between experiments using muscle cells and those using isolated actin and myosin ensembles are not limited to the FV relationship. Thus, a comparison of muscle fiber data [64] to in vitro motility assay data (with actin filaments propelled by surface-adsorbed myosin motor fragments) at a given temperature showed a similar maximum sliding velocity and similar MgATP concentration (K_M_^V^) for half maximum velocity [23,65]. However, admittedly, equally good correspondence was not observed in some other studies [18,66]. It remains to be clarified whether these differences between studies are related to the use of different surface substrates for adsorption of myosin motor fragments (nitrocellulose coated [18,66] vs. trimethylchlorosilane derivatized [23,65] glass cf. [67]) or other factors.

In further support of the current hypothesis, it is worth noting that efforts to estimate a range of single molecule parameters (power-stroke distance, cross-bridge stiffness) from mechanical experiments on muscle cells and single myosin motors interacting with one actin filament, generally, give quite similar values [23].

Finally, strain-dependent transitions, for example, with different actomyosin cross-bridge detachment rates for different distortions of the cross-bridge, are critical in accounting for the steady-state FV relationship according to models from Huxley [44] and onwards. Therefore, it is of interest that quantitative data directly demonstrating such strain dependence can be extracted from interactions between single myosin molecules and one actin filament [68,69].

### 3.3. Evidence Against the Hypothesis

Kaya et al. [37] recently presented data that seemed to argue against the assumption that the single molecule properties could predict ensemble contraction without invoking cooperative phenomena. These data were derived using a setup similar to that described in the legend of Figure 2, with full length myosin molecules co-polymerized with myosin rods into a myosin-rod co-filament, allowing less than 20 myosin molecules to interact with an actin filament held by an optical trap. Displacements were, then, recorded at different load levels. Using this setup, Kaya et al. [37] found 4 nm stepwise actin displacements at a rather high load (>30 pN). Due to the fact that the mechanical work of 4 × 30 pN nm = 120 pN nm ≈ 30 k_B_T is greater than the free energy of MgATP turnover (25 k_B_T, cf. Appendix A), the observation was taken to imply that the steps were not due to single myosins but potentially due to coordinated force generation by several myosin motors. As Kaya et al. [37] stated, “our findings reveal how the properties of skeletal myosin are tuned to perform cooperative force generation for efficient muscle contraction“. In partial contradiction of this statement, one can argue that the findings are actually consistent with the current hypothesis where cooperative effects are deemed to be unimportant. However, in this context, it is extremely important to very clearly define what is meant by cooperative interactions. Obvious cooperative effects would be different kinetic actomyosin properties in ensembles as compared with individual motors, for example, with different strain dependence of transition rates or new transitions in ensembles as compared with the single molecule case. In this case, new transitions could include slippage transitions between neighboring sites on actin and cooperativity between the two myosin heads only in ensembles. Such phenomena were not operative in the studies of Kaya et al. but one could also consider the possibility of cooperativity between different myosin molecules in the ensemble, which seems similar to what was found by Kaya et al. [37]. However, importantly, the latter type of cooperative effect does not prevent conclusions about ensemble contraction based on single molecule properties. Thus, the kinetic properties of other motors (see above) were not altered by the actions of one of them in the work of Kaya et al. [37]. Instead, their findings could be explained by the fact that actions of a given motor changed the distance of the actin filament sites relative to a fixed reference point on the thick filament. This changed various interstate transition rates due to altered strain, but it neither implied a change in the strain dependence or the rates of the individual motor (see also Duke [70]) nor the introduction of new types of transitions. Similar effects have actually emerged in a range of recent cross-bridge models, most notably the bottom-up models [29,31,32,61] mentioned above, which are considered to be strong pieces of evidence in favor of the current hypothesis. Thus, the latter models [29,31,32,61] also gave effects (Figure 4) with several (2–4 under unloaded conditions) coordinated strokes and sub-strokes when implemented using a Monte Carlo approach for similar conditions (~20 myosin motors interacting with one actin filament), as in the work of Kaya et al. [37]. However, the Monte Carlo model, used in Figure 4, gives similar predictions for steady-state FV relationships as the steady-state solution of differential equations in state probabilities [32] if the number of interacting myosin heads is increased towards infinity, consistent with the current hypothesis.

Several mechanokinetic models (see [71,72,73] and references therein) require the assumption of faster attachment rate during shortening, to account for high power output, than required to account for the rate of increase in isometric force following a release imposed under conditions of maximum activation. This has led to schemes that assumed a velocity dependent attachment rate constant [72,73] where the latter was highest at shortening velocities with maximal power output as compared with other velocities. In reality, such a velocity dependent attachment rate constant could correspond to sequential actions of the two myosin heads [59,74,75] or slipping transitions of each individual head between neighboring sites on actin [31,32,76] at certain velocities. Furthermore, [71] considered the possibility that an apparent velocity dependence of the attachment rate was related to thick filament based activation where myosin heads increasingly swung out from the thick filament backbone when the backbone was subjected to increasing tension. 

However, somewhat surprisingly, recent models of the bottom-up type have predicted maximum power output in the experimentally observed range (Figure 2 and Figure 5), assuming the same attachment rate constant as required to account for the maximum rate of increase in force during a tetanus [29,31,32]. Thus, in these models, there does not seem to be any critical need to assume a velocity dependent attachment rate constant to account for high maximal power (Figure 5). The reason for greater success of the recent bottom-up models, in this regard, is not entirely clear. Possibly, it reflects the careful derivation of parameter values from isolated proteins under conditions as similar as possible with regard to ionic strength, temperature, and species [29,31,32] (see further notes of Appendix A and supporting text). It is also suggested, based on the data in Figure 5, that the power output was further increased in models assuming nonlinear cross-bridge elasticity, as found recently in single molecules [77]. Finally, a possible basis for improved prediction of both maximum power output and the rate of increase in isometric force may be that fine details of the models have changed as compared with earlier models. For instance (Figure 6), a rather small change in position of the free-energy minimum, x_1_ of the initial stereo-specifically bound pre-power stroke state with ADP and Pi at the active site (AMD_PP_ in Figure 3A), leads to opposite changes in power output and the predicted rate of increase in isometric force. This can be understood because an increased value of x_1_ increases the distance over which a cross-bridge can develop positive force during sliding, contributing to increased power. At the same time, an increased value of x_1_ makes force development slower because the optimal point of cross-bridge attachment is shifted, to slow the transition into subsequent high-force states if there is no sliding between actin and myosin.

The key regulatory proteins on the thin filaments are troponin and tropomyosin. According to current models of thin filament-based activation [78,79], binding of Ca^2+^ to the troponin C unit of troponin leads to release of the tropomyosin-troponin complex from a linkage via troponin I to the actin filament surface. This in turns leads to a shift in equilibrium position of the troponin-tropomyosin complex on the thin filament surface from what is often termed a blocked position at nM Ca^2+^ (pCa < 7) in relaxation, to what is termed a closed position at µM Ca^2+^ (pCa > 6) in activation. In the latter position, the myosin-binding sites of actin are exposed to allow binding of myosin motors and force generation. Such binding in strong-binding states is believed to lead to a further shift of the tropomyosin-troponin complex over the thin filament surface to a fully open state. The most important contractile manifestation of these events is a sigmoidal increase in force with a decrease in pCa from <7 to about 5. This is a major reason for assuming that force development early during a tetanus or other phenomena at low levels of activation cannot readily be accommodated within the current hypothesis. 

In stating the hypothesis, we assume that troponin and tropomyosin do not appreciably change steady-state contractile properties at full activation. Experimental data seem to cast this statement in doubt to some extent. Thus, it has been consistently found [80,81,82] that reconstitution of actin filaments with troponin and tropomyosin, to produce “regulated thin filaments”, increased the maximum isometric force at full activation (µM Ca^2+^). This increase in force has been attributed to increased force per cross-bridge attributed to strengthened hydrophobic interactions between actin and myosin [83]. In particular, the effect of tropomyosin and troponin reconstitution on force seems to be attributed to tropomyosin [67]. However, reported effects of tropomyosin on velocity seem less consistent. Thus, using what seems to be saturating HMM density on an in vitro motility assay surface, Homsher et al. [66] found higher maximum myosin induced sliding velocity for regulated thin filaments (with both troponin and tropomyosin) as compared with pure actin filaments at full Ca^2+^ activation. A similar effect was, however, not observed by van Buren et al. [84], by adding only tropomyosin to the actin filaments. Moreover, Marston reported [67] that, whereas regulated thin filaments exhibited higher sliding velocity at full Ca^2+^ activation than actin filaments, when propelled by HMM adsorbed to nitrocellulose, (as also used in [66]) no such effect was observed when HMM was adsorbed to silanized surfaces. 

The myosin binding protein C (MyBP-C) is believed to play roles in thick filament activation relying on tension sensing [85,86], as well as sensitization to Ca^2+^ in thin filament-based activation [86,87,88,89]. In addition, however, it has been found, using both skeletal muscle and cardiac muscle proteins, that MyBP-C affects function at full Ca^2+^ activation. Thus, both actin filaments without regulatory proteins and native thin filaments (including both troponin and tropomyosin) that slide along native thick filaments at µM Ca^2+^, exhibited appreciably reduced velocity when they reached the C-zone where the MyBP-C was located [87,88,90]. In line with these findings, Robinett et al. [91] concluded, from their studies of skinned skeletal muscle fibers that “MyBP-C and its phosphorylation state regulate sarcomere contraction by a combination of cross-bridge recruitment, modification of cross-bridge cycling kinetics, and alteration of drag forces that originate in the C-zone”. 

Spatially explicit mechanokinetic models are models that take into account central aspects of the geometrical relationship between thin and thick filaments, and in several cases, include key accessory proteins, as well as the elastic properties of the filaments [14,15,19,92,93]. This gives rich potential to fit details and test intricate predictions involving both geometrical changes and accessory proteins, albeit with certain potential drawbacks [29].

Spatially explicit modeling studies have suggested that the detailed geometry was important to accurately predict certain phenomena. According to one study [14], this included a higher force output per cross-bridge and a higher fraction of attached cross-bridges for a model that explicitly takes into account the 2:1 ratio and the actual geometry between thin and thick filaments, as well as the elastic properties of both filaments and cross-bridges. 

Finally, potential radial components of cross-bridge force have been suggested to be important for function [15,16] as both axial and radial forces may depend on lattice spacing. However, in relation to the latter result, it is of interest to note that experimental data suggested limited changes in these properties upon altered sarcomere length in active muscle fibers if the effects of passive elastic components were taken into account [94,95,96]. The change in sarcomere length tests the role of inter-filament spacing due to constant volume behavior of the myofilament lattice [13] in intact muscle fibers which leads to a reduced inter-filament distance with increased sarcomere length. 

To summarize, there is evidence to indicate that either the maximum isometric force or the maximum velocity of shortening, or both, are affected by the presence of accessory proteins and that models that take 3D order into account may lead to different predictions than simpler models. It is important, in the context of these results, to clarify to what extent such effects are consistent with the evidence presented above, in favor of the current hypothesis. Thus, are any effects of accessory proteins and 3D order at high activation levels so minor that they are not detected when comparing predictions of bottom-up models to experimental data and when comparing experiments from muscle/myofibrils to those on small disordered actomyosin ensembles? In that case, the hypothesis would be approximately valid and highly useful for first order understanding of muscle function. Alternatively, could different experimental systems and conditions using isolated proteins modulate the phenomena studied, as exemplified by effects of surface substrates in the in vitro motility assay [67,97,98]. In any case, additional dedicated experiments are required to more definitely clarify these issues.

### 3.4. Future Tests of Hypothesis

Further testing of the hypothesis will require experimental systems that enable addition/removal of one natural component (e.g., accessory proteins, hierarchical order) at a time and direct studies of elastic properties of myosin motors with intact myofilament lattice. Furthermore, it will be essential to exclude, while using these experimental systems, that the addition/removal of one component does not introduce complicating factors such as new interaction points with artificial surfaces or disorder of the myofilament lattice (Figure 7). 

The following two principles, or combinations of them, may be considered for adding and removing components: (1) Bottom-up assembled systems and (2) top-down disassembled systems (see Introduction).

### 3.5. Bottom-Up Assembled Experimental Systems

In this approach, one starts with purified protein components, for example, MyBP-C, troponin, and tropomyosin, which can self-assemble with myosin and actin filaments, respectively. The simplest examples are unloaded in vitro motility assays where different components such as troponin, tropomyosin, and MyBP-C are added, and bind to actin and myosin, followed by studies using conventional motility assays to elucidate effects on maximum unloaded sliding velocity [84,100,101,102]. However, it is of interest to extend these assays to systems that allow detailed studies of the force-velocity relationship. Whereas such studies are possible to some extent using so-called loaded in vitro motility assays [80], the latter are difficult to use for detailed quantitative analysis of the FV relationship and power output. 

Particularly, assays of the type illustrated in Figure 7 (cf. [22,36,37,63]), where force and displacement are recorded for small ensembles of myosin motors, would be useful for testing the present hypothesis. They would allow studies of the FV relationship upon addition/removal of different protein components, one after the other, under otherwise similar conditions. After supplementation with microfluidics methods, for rapid solution exchange (cf. [103]) or the use of caged compounds [104,105], similar experimental systems may be useful to study tension transients in response to rapid concentration changes of, for example, Ca^2+^, ATP, or inorganic phosphate. Possible modifications, as compared with recently published work [22,36,37,63], that may be of interest is the introduction of microscale square pedestals for motor adsorption, rather than curved surface which has been used in recent optical tweezers studies [36,37]. This would allow better control of the number of motors, facilitate production of various ordered arrays of myosin motors, and combination with total internal reflection fluorescence (TIRF) microscopy. The latter could be used for studies of ATP turnover by single molecule techniques [106,107] and single molecule fluorescence resonance transfer (FRET) for studies of structural dynamics within the myosin motor. With regard to the production of ordered arrays of myosin motors, DNA nanotechnology (DNA origami) [108] has some advantages, for example, to produce unipolar arrays. However, it is important to be able to compare the results with those where myosin motors are incorporated into native thick filaments. An experimental system studying a large number of motors (>> 20) would enable more detailed characterization of the force-velocity relationship by reducing force fluctuations (e.g., related to inter-motor cooperativity in very small ensembles) [37] and achieving a situation with a large number of available cross-bridges more similar to muscle cells. This would, however, require a nano/micro-fabricated cantilever-based force measuring system [22,99] rather than optical tweezers, and possible problems related to thin filament stability at high forces would need to be addressed. What would be of appreciable interest is if parts of the 3D order of the sarcomere could be reconstructed in a system similar to that in Figure 7, for example, using scaffolds created by combinations of top-down (e.g., lithographic techniques) [43,109] and bottom-up (DNA origami) [108] nanotechnology.

As with all experimental interventions, bottom-up studies come with a number of challenges that must be circumvented to avoid non-physiological results. This includes challenges related to protein production, whether by purification from tissue, for example, using proteolytic enzymes or by protein expression. In this process, parts of the molecule could, at least to some degree, be cleaved off [90], be partly unfolded, as is a challenge with expression of striated muscle myosin II [110], or lack important posttranslational modifications [111]. A further challenge, in experiments with bottom-up reconstruction, is to ensure that the reassembly with myosin and actin leads to physiologically relevant interactions between all proteins involved. Particular concerns may arise in this regard when using fragments, rather than full length proteins, of for example, MyBP-C [86,87]. Additionally, as touched on briefly above, it is important that the re-assembly or the whole artificial setup does not introduce complications, for example, due to unwanted surface-protein interactions [67]. 

### 3.6. Top-Down Disassembled Systems 

Myofibrils isolated from muscle represent one hierarchical level of disassembly. However, the myofibrils are still highly complex structures, containing most sarcomere proteins (that are not soluble) in addition to actin and myosin. Therefore, further disassembly to investigate the effects of specific protein components, by their selective removal, should be considered in a context with similar efforts using skinned, demembranated muscle cells. Nevertheless, further disassembly to selectively remove thick filaments from just preselected sarcomeres, for example, to investigate the effects on inter-sarcomere dynamics, is greatly facilitated by the use of myofibrils [112,113]. However, in studying the role of specific sarcomere proteins, such as troponin, tropomyosin, MyBP-C, and titin, achieving the selective removal/addition of these is of interest, more generally for all sarcomeres, whether in myofibrils or muscle cells.

The top-down disassembly of sarcomere components can take the simple form of removal of specific wild-type protein components (e.g., troponin C) [114], possibly followed by reconstitution with a modified protein.

In contrast, an appreciably more extensive disassembly can be achieved using the actin severing protein gelsolin to remove the thin filaments [81,83,115] or a solution of increased ionic strength to remove thick filaments. In the case of the thin filaments, the disassembly can be followed by bottom-up reconstitution by addition of the thin filament components, i.e., G-actin, tropomyosin, and troponin, opening for a wide range of studies [81,83,115]. The latter includes effects of the different proteins on force and motion generation (see above) but also the effects of mutations in, for example, troponin and tropomyosin [116]. To the best of my knowledge, no similar reconstitution approach as for the thin filaments, is available for thick filaments. 

The selective removal/addition of proteins while maintaining sarcomere structure has appreciable potential for investigating effects of the protein on the ordered filament network. However, this approach may be prone to complexities. This includes uncertainties about the fraction of successfully removed normal proteins, as well as quantification of the successful fractional reconstitution with proteins (e.g., troponin/tropomyosin/actin stoichiometry). Furthermore, it is essential that the procedure to remove/add protein components does not change other potentially important characteristics of the contractile system, such as the degree of myofilament order or inter-sarcomere uniformity.

A type of disassembly that may be viewed as appreciably more radical than selective removal of individual protein components from otherwise intact sarcomeres, is the isolation of native thick and thin filaments from muscle. This procedure has the potential to provide other types of valuable information than studies using the bottom-up (reconstituted filaments) approach [90,117]. However, one must keep in mind that such native filaments are also modified to different degrees during the purification process. For instance, the titin link between the thin and thick filaments in the sarcomere is removed and there may also be other effects, for example, due to the use of proteolytic enzymes [90]. Experimental challenges also include the limited length of the filaments and the bipolar nature of thick filaments with myosin heads extending in different directions on either side of the M-line. These effects complicate mechanical studies, for example, manipulating the filaments in force-measuring setups (cf. Figure 7).

### 3.7. The Cross-Bridge Elasticity with an Intact Myofilament Lattice

As pointed out above, the elastic properties of the actomyosin cross-bridges are of importance for some phenomena related to the proposed hypothesis. Whereas these properties may be determined in assays of the type illustrated in Figure 7, this is not of significant interest because it is quite generally accepted that the elasticity of isolated myosin molecules is nonlinear [77]. The nonlinearity has instead been questioned, particularly for preparations such as myofibrils and muscle cells with an intact 3D ordered myofilament lattice. A method to characterize the elasticity in such a system was proposed recently by [61]. Thus, nanometer tracking methods [65,118] are suggested for probing the distance between fluorescent molecules/particles (e.g., quantum dots) on the myosin filament backbone and the myosin motor domain (e.g., attached via a regulatory light chain) in myofibrillar preparations.

## 4. Conclusions

A hypothesis has been presented, together with evidence that corroborates it, as well as pieces of evidence that argue against the hypothesis. Further studies, such as suggested above, are required. Most likely, considering the potential complexities involved, the issue cannot be fully resolved without the combined use of several types of bottom-up and top-down experimental approaches. Currently, in this paper, the opinion is that the hypothesis is true to a high degree of approximation. However, the hypothesis is highly useful, even if a weaker form of the hypothesis could be corroborated and if the phenomena that it does or does not apply to could be more clearly delineated For the conditions where it is valid, it would avoid the uncertainties of assigning parameter values and interpreting more complex models that would have to take into account both effects of accessory proteins and 3D order. It may also be important for in depth insight into a range of phenomena to better clarify under which specific conditions the hypothesis best approximates contractile function. For instance, if the hypothesis is a very good approximation for normal muscle proteins but not for proteins modified by diseases (e.g., in HCM as suggested previously [23]) this might give clues to disease mechanisms. Such knowledge is also of importance for drug discovery efforts by clarifying to what degree experimental findings using isolated proteins can be extrapolated to the tissue level. In this paper, the validity of the hypothesis has been tentatively limited to high levels of activation. However, the limitations need to be delineated in greater detail as well as their possible change under disease conditions.

## Figures and Tables

**Figure 1 ijms-21-08399-f001:**
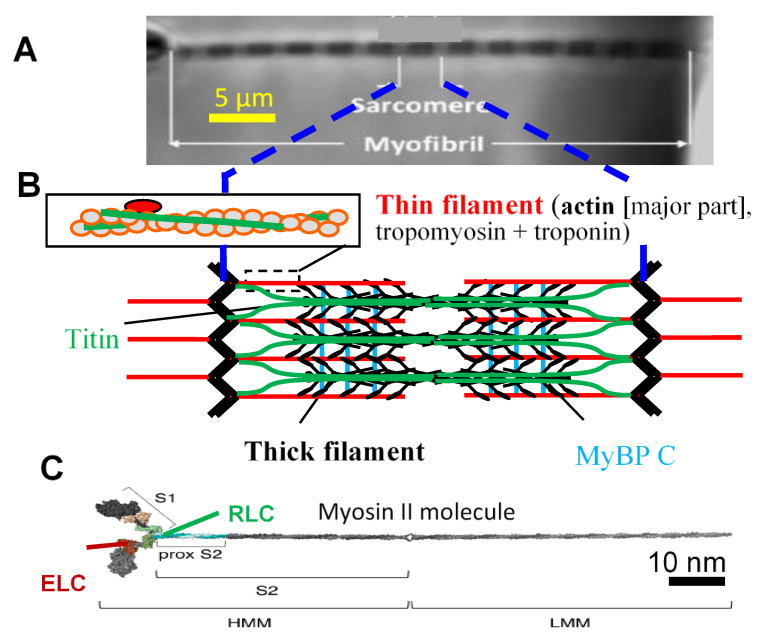
Hierarchical organization of muscle. (**A**) Isolated myofibril mounted for force measurements [20]. Myofibrils, with repeating structures, sarcomeres, fill the muscle cells in a highly ordered arrangement; (**B**) Schematic illustration of sarcomere (longitudinal view), with key protein components, i.e., thin filaments with actin and regulatory proteins and thick filaments with myosin heads (cross-bridges, black) extending from myosin backbone and accessory proteins, i.e., titin and myosin binding protein C (MyBPC); (**C**) Isolated full-length myosin molecule with two motor domains (left, S1, heads). Essential (ELC) and regulatory (RLC) light chains are attached to the lever arm of each of these domains. Panel C modified from [21] under license CC BY 4.0.

**Figure 2 ijms-21-08399-f002:**
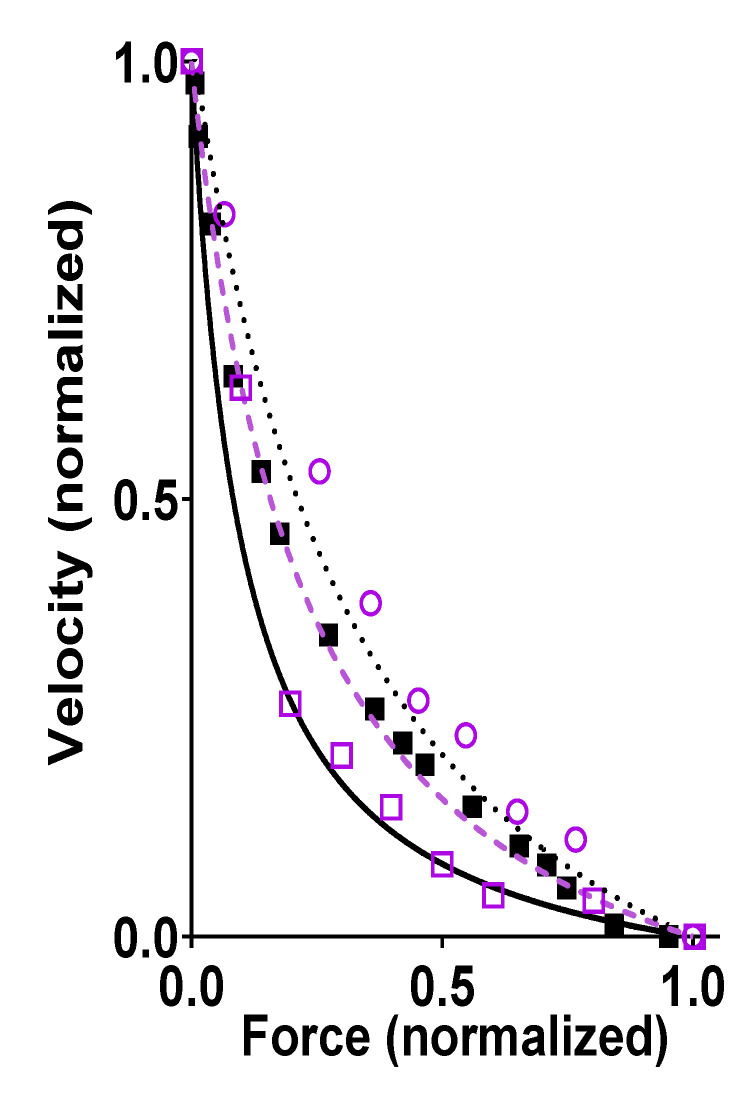
Force-velocity (FV) relationship, i.e., velocity of shortening of fast mammalian skeletal muscle vs. developed force. Data recorded from living fast skeletal muscle (black) of mouse [38] (full squares), rabbit (extraocular muscle, full line) [39], or rat (dotted line) [40]. Alternatively, data were obtained (purple) from an isolated ensemble of fast rabbit myosin interacting with a single actin filament including data from Pertici et al. [36] (dashed line), Kaya et al. [37] (open circles), and Cheng et al. [22] (open squares). The ensemble data for isolated proteins were derived by either first adsorbing myosin motor fragments (heavy meromyosin) to optical fiber surfaces [36] (~<20 myosin heads) or by using filaments (native thick filaments [22] (~<100 myosin heads) or myosin-rod cofilaments [37] (~<20 myosin heads)) with full length myosin, and then bringing an actin filament, held by an optical trap [36,37] or by a cantilever [22], in contact with the motor ensemble to record force and displacements. Force and velocity in this figure are normalized to their maximum values. Data obtained by measurements from figures in the referenced papers (when symbols are shown) or by reproducing the Hill hyperbolic function (when lines are shown) fitted to the data in the original papers.

**Figure 3 ijms-21-08399-f003:**
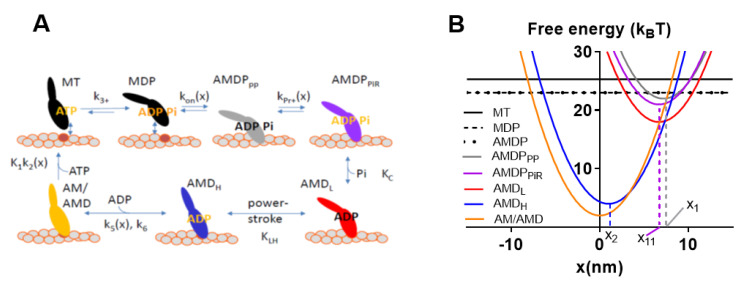
Model [32] slightly modified as in [30] used for simulation of data for Figures 4 and 6 below. (**A**) Kinetic scheme with labeling of different myosin (M) states according to actin-binding (AM) and substrate (ATP:T) or products of ATP turnover (ADP:D, inorganic phosphate P/Pi) at the active site. Numerical values of equilibrium constants (uppercase letters) and rate constants (lowercase letters) given and motivated in Appendix A and associated supporting text. Subscripts of the different states have meanings as follows: PP, prepower stroke state; PiR, Pi-release state; L, low-force strongly bound state; H, high-force strongly bound state; (**B**) Free energy diagrams for the different states in (A) (with same color codes and labeling). The parameters x_1_, x_11_, and x_2_, indicated in the figure, denote the x-values where the free energy of the states AMDP_PP_, AMDP_PiR_ (as well as AMD_L_), and AMD_H_, respectively, attain their minimum value.

**Figure 4 ijms-21-08399-f004:**
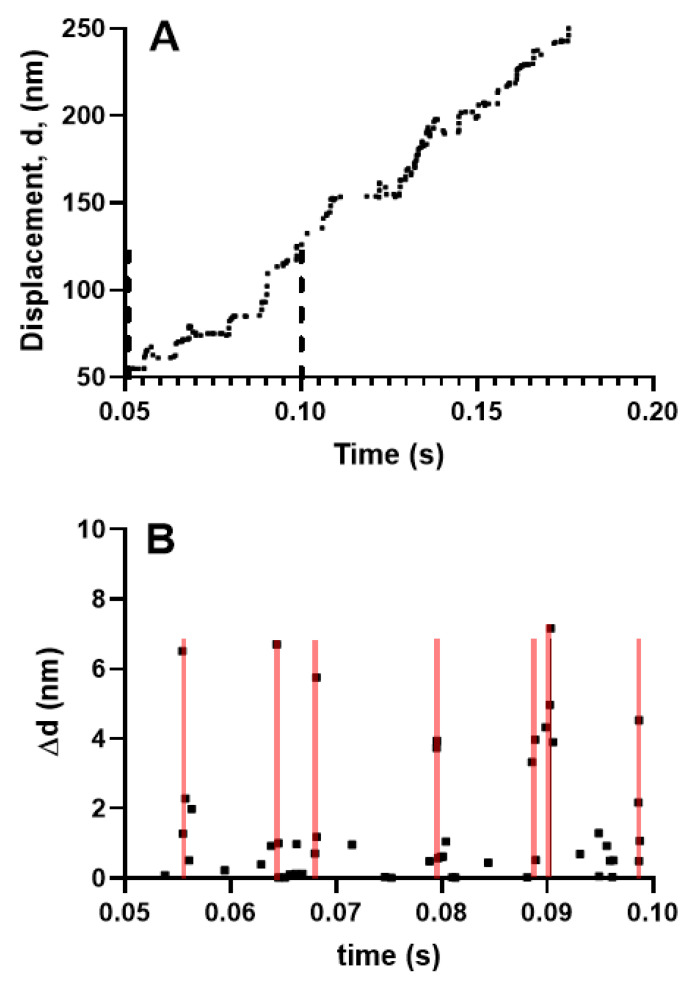
Simulated myosin propelled actin filament sliding produced by 18 myosin molecules against zero load. This number of molecules is similar to that used in the experiments of Kaya and Higuchi (2017) [37]. (**A**) Displacement record, time interval between vertical dashed lines considered in greater detail in (B); (**B**) Step changes in displacement, ∆d from region within dashed lines in A. Each point represents a step change in displacement. Note that substantial fractions of the steps are clustered in seven different 0.4 ms time intervals (covered by transparent red lines) with 2–4 steps, of amplitudes from ~1 nm to ~7 nm in each cluster.

**Figure 5 ijms-21-08399-f005:**
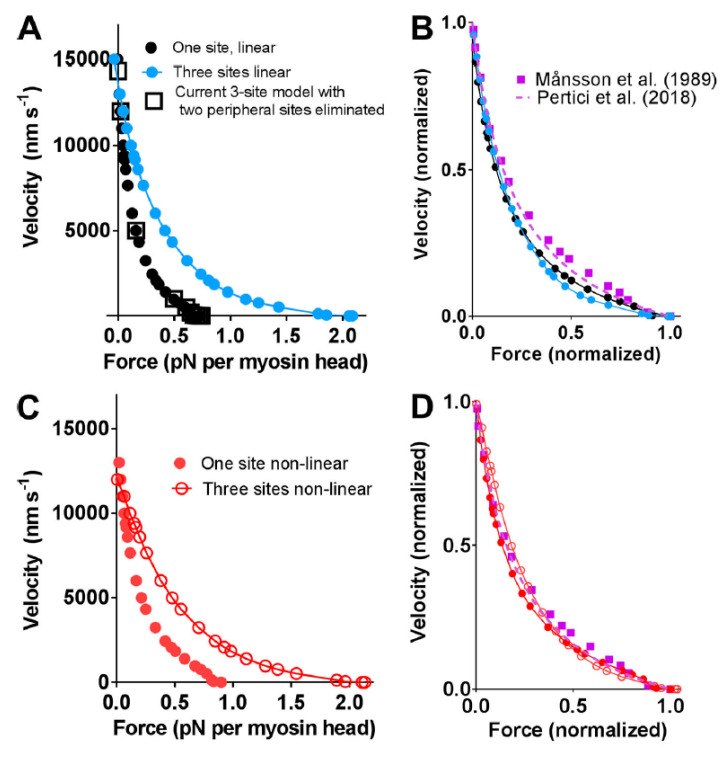
Force-velocity relationships predicted by bottom-up types of models assuming one or three binding sites for myosin on neighboring actin subunits in target zones separated by 36 nm along the actin filament as compared with experimental data from Figure 2. (**A**,**B**) Linear cross-bridge elasticity with stiffness 2.8 pN/nm assumed; (**C**,**D**) Nonlinear cross-bridge elasticity similar to that found by Kaya and Higuchi in single molecule studies. Figure reproduced from [29] under license (CC BY-NC-SA 4.0). Model similar to that described in Figure 3 but without the initial pre-power stroke state (AMADP_PP_) that was introduced [32] to account for the effect of the small molecular compound blebbistatin (see further Introduction) and with slightly modified parameter values (within experimental uncertainties, details in [29]).

**Figure 6 ijms-21-08399-f006:**
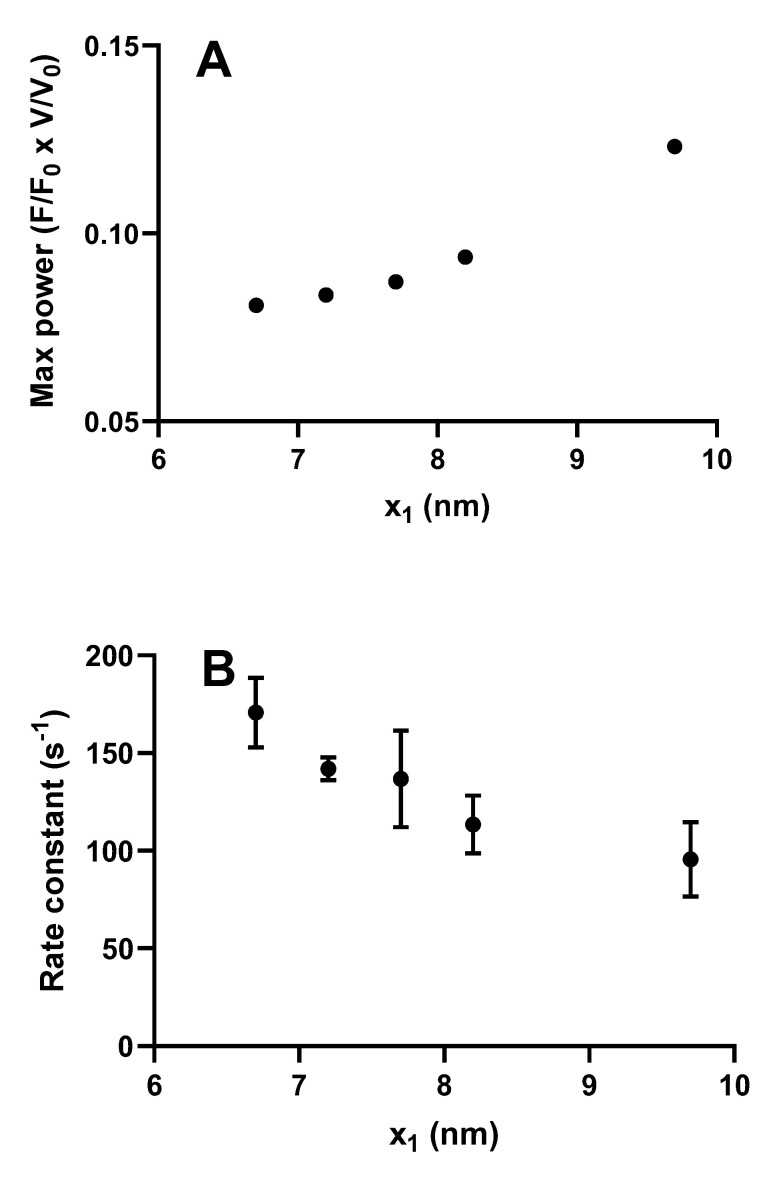
Maximum power and rate of increase in force during an isometric contraction simulated for different values of the parameter x_1_ in the model described in Figure 3 (similar to that in [32]). (**A**) Maximum power derived by steady-state solutions of differential equations in state probabilities. Statistically significant (*p* = 0.0167) correlation (Spearman’s r = 1) between x_1_ and maximum power; (**B**) Rate of increase in force starting from all myosin heads in the AMDP_PP_ state estimated from fits to 4 Monte Carlo simulations at each x_1_ value. In each of these cases, the parameter x_11_ was changed in parallel with x_1_ with x_11_ = x_1_ −0.5 nm. Data are given as mean ± standard error of the mean. Statistically significant (*p* = 0.0010) correlation (Spearman´s r = −0.68) between x_1_ and rate constant.

**Figure 7 ijms-21-08399-f007:**
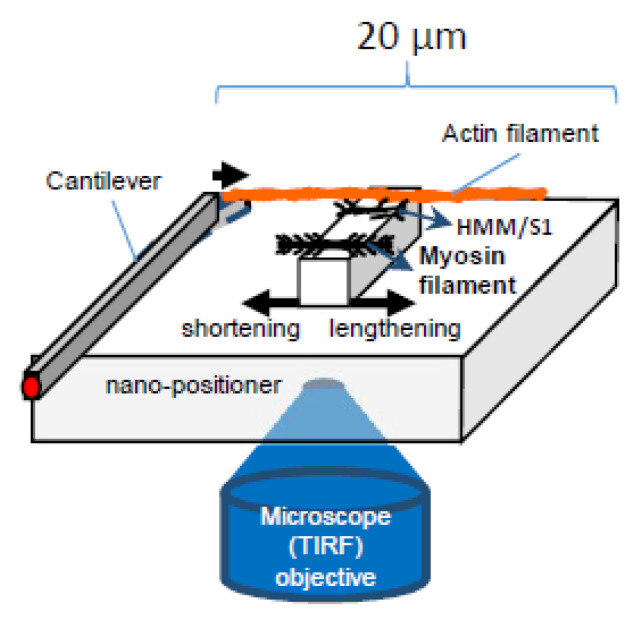
Schematic of experimental system to study effects of force and velocity, building on previous work [22,36,37,99].

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
