# Peer review of "Hypothesis: Single Actomyosin Properties Account for Ensemble Behavior in Active Muscle Shortening and Isometric Contraction"

_ijms, 2020, doi:10.3390/ijms21218399_

Round 1

Reviewer 1 Report

In this ‘hypothesis’ paper the author suggests that the contractile behaviour of intact muscle can be pretty well explained, under defined circumstances, by looking at the properties of model systems involving relatively few molecules.  Various observations from intact muscle are claimed to be modelled quite well using parameters derived from laser trap and other studies of small molecular assemblies.  

The hypothesis is put forward that ‘Single molecule properties of actin and myosin account for ensemble contractile properties of a half-sarcomere during shortening and isometric contraction at nearly saturating Ca-concentrations.’

The paper ends with a discussion of features that appear not to be explained by the model systems and then it suggests experiments that might further test the validity of the hypothesis.

In the past I have not been directly involved with calculations such as those in this paper, so I have to take much at its face value.     However, in my own studies I am very concerned that when modelling is done against unknown parameters, the number of parameters must be much less than the effective number of observations for any analysis to be considered reliable.     If this is not the case the problem is indeterminate and there will be many equally valid solutions.    Perhaps the author could say whether the calculations in the paper are based entirely on parameters obtained directly from previous experimental studies, and if not, how many free, unknown, parameters are being used to refine the fit to the observations.     Are there more observations than free parameters?

In more detail, on page 4 (top half) the author states that 556 myosin heads were used.     Where does this number come from?     Considering one vertebrate striated muscle myosin filament surrounded by six actin filaments, there are 294 myosin heads available to interact with actin (49 x 6) in one half A-band.     There are two actin filaments per myosin filament so there will be 147 heads available to interact with one actin filament.      Most estimates of the number of heads attached in strong states to actin in full overlap isometric contractions give a value of around 30% attachment.     So there will be on average roughly 0.3 x 147 = 44 heads in strong states on any one actin filament.   At full overlap, the overlapped part of actin is around 7140 Å (or 259 actin monomers; 518 for two filaments), which is roughly 20 actin repeats of around 360 Å per filament.    So in full isometric contractions there will be roughly 2 to 3 (44/20) myosin heads binding per actin target area repeat.     556 seems a strange number to use!    And to quote: ‘Therefore, all myosin heads (corresponding to a saturating density on a surface) are distributed uniformly in the simulations between 360 bins each of 0.1 nm width’.       How can that correspond to heads periodically placed in actin target areas?  In short, I do not understand the premiss of the calculations.

If the author can show that the basis of the calculations is reasonable, and also that there is not an excess of free parameters over observations, then I think this exercise is a useful one.    To me it is of fundamental importance in the muscle research field at the present time to find the simplest possible explanations for various muscle properties.    This is the principle of Occam’s razor; we need to know what is fundamental and what is not!

The English can be improved in places:

Line 38:   evolutionarily

Line 56:  longitudinal view (not section)

Line 87:   ‘state the following hypothesis’ should probably not be in italics

Line 88: (and elsewhere):     The single molecule……

Line 273:    change not changes

Line 340:   increases no increase

Line 348:   A similar effect……

Line 377:     tests the role (not tests of the role)

Line 428:  has not have

Line 429:   results with those (not to those)

Line 508:  remove that

Author Response

I am grateful for insightful and constructive reviews of the manuscript that have
helped me to achieve what, I believe, are major improvements. Please find below
my detailed point to point response to the reviewer comments with the original
comments in italics. In addition, the key changes are also indicated by highlighted
text in a submitted version of the manuscript for review only. The Supporting
Information has also been extensively updated to the extent that it was not
meaningful to indicate the changes by highlights in the text. This encompasses
extensive changes to the Tables S1 and S2 and addition of a new Supporting Text
section. Please note, that a list of references that I cite below are located after the
responses to both reviewers.

Response and changes made: First of all, it must be stated that all parameter values
are fixed to start with and whenever possible they have been derived directly from
2 (11)
independent experimental data using isolated proteins and, to limited extent (indicated in
Tables S1-S2 from independent skinned fiber experiments. Finally, in some cases the
model parameters have been determined in less direct ways when explicit quantitative
data are not available, e.g. based on semi-quantitative data and supporting
argumentation. How specific parameters have been obtained has been described
previously. However, I realize now that it is NOT a good idea to force readers of this
paper to go to those earlier sources for information. Furthermore, I also see that it is
most fair, when putting forward a somewhat controversial hypothesis, to give all details
in the same context. Now, I have therefore added information to the Supporting Tables
with references to the papers from which the parameter values originate. Additionally, I
have added Supporting Text that describes the origin of the parameter values also in
cases when it is difficult to find results in the literature from single molecules or solution
biochemistry. In these cases I have, now (in the new Supporting Text) in greater detail,
than in previous studies described the argumentation that have led to the actual
parameter values used.
Importantly, fixing of the parameter values in efforts to predict experimental findings (no
free parameters to be fitted) generally gives sufficiently good predictions to be within the
experimental uncertainties (including ubiquitous variations between labs; present Fig. 2
as an example), requiring no further adjustment of the parameter values. However, in
some cases, particularly when the aim has been to accommodate specific data from a
given lab, the model parameter values have been adjusted within a 20 % experimental
uncertainty, originating in the experiments (usually single molecule studies or solution
biochemistry) where they were determined. In some cases, the adjustments have been
more extensive when there has been appreciable uncertainty in a parameter value. This
was the case for the recent updates1 of kp+´, x1 and x11 to their present values from
those used initially2
. All this has now been described in detail in the updated Supporting
Tables and associated Supporting Text.

I am sorry about these confusions. I fully understand the reaction to the
number of 556 myosin heads and I now immediately see that it is quite impossible to
understand where it comes from if one does not (in some detail) study the Supporting
Information of the papers which I refer to. (Again, sorry about this). The basis for the
number of 556 is that I have made the following simplifying assumptions, related to the
origin of muscle function in single actomyosin properties: 1. A muscle half-sarcomere
behavior can be approximated by the behavior of large number of surface adsorbed
myosin motors interacting with a sufficiently long single actin filament (here taken as 20
µm) in an in vitro motility assay configuration (with the added possibility to vary load on
the filaments or velocity). 2. On these premises the surface density of active myosin
motors can be varied from very low to simulate the experimental data of Kaya et al
(present Fig. 4) to high for simulating the properties of a muscle half-sarcomere. 3. The
total number of myosin binding sites along a 20 µm long filament is 20*1000/36≈556 on
the assumption of 36 nm distance between neighboring sites.

The situation from the in vitro motility assay mentioned above is easily transferred to the
situation of a muscle fiber by assuming that we look at twenty 1 µm long (thin) actin
filaments (with 556 binding sites for myosin) interacting with ten half (thick) myosin
filaments. If each thick filament contains 294 myosin heads there would be 147 heads
per thin (actin) filament. This would be very similar to what is achieved in the in vitro
motility assay approximation in the simulations. Assuming a saturating surface density of
5000 active myosin heads per µm2
, a 20 µm long filament and a band of 30 nm width
around the filament where myosin heads may reach actin on arrives at a total number of
heads of: 5000*0.03*20=3000 per 20 µm of an actin filament, i.e. 150/µm. However,
notwithstanding the latter similarity to the value of 147 heads per actin filament in
muscle, the force is anyway easily normalized to the force per number of available
heads. Furthermore, clearly (under the current hypothesis), the force and velocity is the
same for a long single filament with a uniform distribution of myosin heads relative to the
binding site as for one single actin filament interacting with a similar number of uniformly
distributed myosin motors. The assumption that the position of the myosin motors in
muscle are uniformly distributed relative to the nearest actin binding site was originally
proposed by Huxley due to the mismatch of the actin and myosin periodicities. To this,
one may add conditions in a muscle (when considering a very large number of halfsarcomeres working in parallel) that there are variabilities in overlap and register
between several of these sarcomeres (see references in new text). Thus, I think that the
idea of a uniform distance distribution is a fair approximation when the very large
number of interacting myosin motors in a muscle are considered. Naturally, however,
this approximation is not valid when the interaction between just very few (say 1-5) thin
and thick filaments are considered explicitly. Finally, it is important to note that the
assumption of uniform distributions are also appropriate in Monte-Carlo simulations of in
vitro motility assays at both high and very low motor densities. However, in this case the
exact position of the limited number of myosin heads relative to the binding sites are
determined by a random number generator that distributes the individual heads in bins in
the range 0 – 36/2 nm to a defined center of the nearest binding site.
Changes made: I have now extensively updated the text in the Methods section (lines
133-191) with the key elements of the above description of the procedure for the MonteCarlo simulations. I have also added some more detailed pieces of information beyond
this specific issue and hope that it makes the modelling approach sufficiently clear for
the present purpose and quite limited use of the modelling. Finally, I have added text to
the Introduction (lines 75-96) to introduce the types of models used.

: I hope that the changes to the manuscript as outlined
in response to the above points fulfill this, not the least the addition of the new
Supporting Text and the inclusion of references in the Supporting Tables where model
parameter values are given.
The English can be improved in places:
Response and changes made:
All the suggested change below have been implemented at positions in the current
manuscript indicated under each point
Line 38: evolutionarily
Changed on current line 38.
Line 56: longitudinal view (not section)
Changed in legend (line 56) of Fig. 1.
Line 87: ‘state the following hypothesis’ should probably not be in italics
Changed on current line 105.
Line 88: (and elsewhere): The single molecule……
Changed on current lines 106, 23, 333 etc.
Line 273: change not changes
Changed on current line 357.
Line 340: increases no increase
Changed on current line 436.
Line 348: A similar effect……
Changed on current line 444.
Line 377: tests the role (not tests of the role)
Changed on current line 473.
Line 428: has not have
Changed on current line 524.
Line 429: results with those (not to those)
Changed on current line 525.
5 (11)
Line 508: remove that
Changed on current line 604.

Reviewer 2 Report

This is a potentially interesting paper, hypothesizing that muscle properties are defined by just actomyosin interaction with no inclusion of other sarcomeric proteins. Then, proteins, other than actin and myosin, form a 3D structure of sarcomere, in which actin and myosin cyclically work and produce force. There are regulatory proteins, but at high calcium, the thin filament regulation is off. The author mentioned MyBP-C regulation (lines 353-3620), but this discussion is not conclusive in the manuscript. The major problem of the manuscript is that in the current form it was written for the narrow circle of actomyosin aficionado, knowing all current literature by heart. To make the manuscript interesting for the general readership the author should include simple two-three sentence descriptions of models and results of referenced papers. I would like to show the good and the bad examples to illustrate the point. A good example, lines 324-327, where Ca regulation is introduced. Or line 207, where the in vitro motility assay is explained. The bad example, lines 257-260, where the results of Kaya et al are not introduced but criticized. Or in the caption to Figure 2 when FV data obtained for isolated actomyosin systems are mentioned, but to understand what these systems are, the reader needs to retrieve the original papers.

In particular:

Line 47, maybe capitalize each word in the Myosin Binding Protein C?

Figure 1 C, fix the typo, Myosin II molecule

Line 59, the caption of Figure 1, C. Myosin is a dimer, it is not correct to say that myosin is a molecule with two heads.

Line 73, “recent modelling work”, performed by the author, should be shortly described in a few sentences, the major feature of the model(s) should be provided. The author spends the whole paragraph telling how good the modeling is, and how the model brings the hypothesis, which is the major point of the manuscript but says nothing about the main features of the model.

Line 102, what is the “isolated ensemble of fast rabbit myosin”? Could the author provide a description which does not require to read the original paper?

Line 108, why the reference of figures 5 and 6 precedes the references of figures 3 and 4?

Lines 108-109, describe in a few sentences the discussed models and parameters, to help the reader understand the models. The sentence in line 111 “for further details, see the original paper” sounds awkward since there are no even basic details in the manuscript.

Line 112, Please define “contractile properties”. If they are obtained from the solution of differential equations, I guess these are the reaction rates. Please indicate what are those rates and how they related to the contractile properties.

Line 113, “Monta-Carlo simulations” of what? A Monte-Carlo simulation means assigning multiple values to an uncertain variable of an equation to achieve multiple results. What are those variables? What is the equation with the uncertain variable?

Line 114, “556 myosin-binding sites on actin”, probably on a thin filament? Could the author provide a reference for the number?

Lines 116-118, Need a better description of a model. These sentences do not have much sense without reading the original paper. If the author discusses the model, why not describe it for a reader?

Line 118, what is the x-value? Is it a position of myosin head on actin filament? What is the physical meaning of x=0? How the x-value is related to energy?

Line 173, Why Figure 4 is on line 173 and the first mention of Figure 4 in a text is on line 297?

Line 173, Figure 4, and the caption. What are the one site and the three sites? Why blebbistatin is mentioned in the caption, and never in the text? Please be consistent with abbreviations, either AMADPpp or AMDpp (line 309). Bottom-up models should be introduced, and their difference explained.  Figure 4 B and D are not clear.

Line 183, please introduce the bottom-up models for the reader, they are not defined in the manuscript.

Line 185, explain what means “three sites per actin target zone”.

Lines 189-191, there is a discussion of models and experimental data and conclusions, but since nothing is introduced, the statement within this text is for the author only, not for the reader. Please fix that.

Line 195, define “small isolated actomyosin ensembles”

Line 250, Figure 5 A, the vertical dashed line at time=0 must be more pronounced, it is barely visible. The term “area” is misleading here. Why show times larger than 0.1s if they are not discussed? What means “appearance of clusters of 2-4 steps within 0.4 ms”? it is not clear from the caption and the text (lines 270-285).

Line 275, a short description of Kaya’s data is needed, otherwise, text in lines 275-285 does not have much sense.

Line 302, define “coherent” conditions

Line 378, the “constant volume behavior” is probably not the right term

Line 437, according to the definition above, the use of lithographic techniques cannot be a top-down approach.

Lines 508-509, please check grammar

Author Response

I am grateful for insightful and constructive reviews of the manuscript that have
helped me to achieve what, I believe, are major improvements. Please find below
my detailed point to point response to the reviewer comments with the original
comments in italics. In addition, the key changes are also indicated by highlighted
text in a submitted version of the manuscript for review only. The Supporting
Information has also been extensively updated to the extent that it was not
meaningful to indicate the changes by highlights in the text. This encompasses
extensive changes to the Tables S1 and S2 and addition of a new Supporting Text
section. Please note, that a list of references that I cite below are located after the
responses to both reviewers.

Response and changes made: These are very valuable comments. Particularly, I
realize that I should have given appreciably more detailed information about models
used as well as fully introduced some central issues. Therefore, I have now
1. appreciably expanded the description of the model by expanded Methods section and
Supporting Information (see further details below).
2. Introduced key concepts and results before their use such as (further details below in
response to specific comments)
-the type of mechanokinetic models used (lines 75-89)
-bottom-up and top-down approaches, more specifically bottom-up models (lines 90-96)
-small ensemble mechanical studies, particularly those of Kaya et al (2017) in legend of
Fig. 2, on lines 274-279 and 335-342.

Specific comments and responses to reviewer 2 below
Line 47, maybe capitalize each word in the Myosin Binding Protein C?
Response and changes made: This has now been changed throughout the manuscript
Figure 1 C, fix the typo, Myosin II molecule
Response and changes made: This has now been corrected in Figure 1C.
Line 59, the caption of Figure 1, C. Myosin is a dimer, it is not correct to say that myosin
is a molecule with two heads.
Response and changes made: Whereas I agree in principle I believe that the
terminology of “myosin molecule” is generally used for the myosin heterohexamer with
two heavy chains and 4 light chains. However, I have now changed to “full-length
myosin molecule” as used in the reference by Spudich and co-workers from which this
picture was adapted.
7 (11)
Line 73, “recent modelling work”, performed by the author, should be shortly described
in a few sentences, the major feature of the model(s) should be provided. The author
spends the whole paragraph telling how good the modeling is, and how the model brings
the hypothesis, which is the major point of the manuscript but says nothing about the
main features of the model.
Response and changes made: The key features of the types of models used have now
been introduced in new text on lines 75-89.
Line 102, what is the “isolated ensemble of fast rabbit myosin”? Could the author provide
a description which does not require to read the original paper?
Response and changes made: This has now been described in detail on lines 274-279
in the legend of Fig. 2.
Line 108, why the reference of figures 5 and 6 precedes the references of figures 3 and
4?
Response and changes made: The reference to figures 5 and 6 at this point has now
been omitted (current lines 133-136).
Lines 108-109, describe in a few sentences the discussed models and parameters, to
help the reader understand the models. The sentence in line 111 “for further details, see
the original paper” sounds awkward since there are no even basic details in the
manuscript.
Response and changes made: In response to this comment, and in response to the
other reviewer, the Methods section has now been appreciably expanded along with the
Supporting information (Tables S1, S2 and new Supporting Text). Additionally, more
details on modelling in general is given in the Introduction section as further described
above.
Line 112, Please define “contractile properties”. If they are obtained from the solution of
differential equations, I guess these are the reaction rates. Please indicate what are
those rates and how they related to the contractile properties.
Response and changes made: This has now been defined in greater detail on lines
137-139 within the expanded Methods section.
Line 113, “Monta-Carlo simulations” of what? A Monte-Carlo simulation means assigning
multiple values to an uncertain variable of an equation to achieve multiple results. What
are those variables? What is the equation with the uncertain variable?
Response and changes made: I hope to have addressed these concerns by the
extensive changes of the text on lines 150-154 within the expanded Methods section
(see also new Supporting Text).